# Impact of Mechanical Stress and Nitrogen Doping on the Defect Distribution in the Initial Stage of the 4H-SiC PVT Growth Process

**DOI:** 10.3390/ma15051897

**Published:** 2022-03-03

**Authors:** Johannes Steiner, Peter J. Wellmann

**Affiliations:** Crystal Growth Lab, Materials Department 6 (i-MEET), Friedrich-Alexander Universität Erlangen-Nürnberg (FAU), 91058 Erlangen, Germany; johannes.steiner@fau.de

**Keywords:** silicon carbide, dislocation networks, numerical simulation, doping, crystal growth, PVT

## Abstract

Nitrogen incorporation changes the lattice spacing of SiC and can therefore lead to stress during physical vapor transport (PVT). The impact of the nitrogen-doping concentration during the initial phase of PVT growth of 4H-SiC was investigated using molten potassium hydroxide (KOH) etching, and the doping concentration and stress was detected by Raman spectroscopy. The change in the coefficient of thermal expansion (CTE) caused by the variation of nitrogen doping was implemented into a numerical model to quantitatively determine the stress induced during and after the crystal growth. Furthermore, the influence of mechanical stress related to the seed-mounting method was studied. To achieve this, four 100 mm diameter 4H-SiC crystals were grown with different nitrogen-doping distributions and seed-mounting strategies. It was found that the altered CTE plays a major role in the types and density of defect present in the grown crystal. While the mounting method led to increased stress in the initial seeding phase, the overall stress induced by inhomogeneous nitrogen doping is orders of magnitude higher.

## 1. Introduction

In recent years, silicon carbide (SiC) has matured as a wide bandgap semiconductor for use in high performance power electronics [1,2,3,4,5]. While defects in the material, such as micropipes (MP), threading screw dislocations (TSD), threading edge dislocations (TED), or basal plane dislocations (BPD), have either been eliminated or greatly reduced; the exact mechanisms of defect generation are still subject to current research [6,7,8,9,10,11,12,13]. SiC is grown using the physical vapor transport (PVT) method, where a high-quality single-crystalline seed is required for achieving a sufficient crystal quality. The growth conditions during the initial seeding phase play a major role in the resulting defect densities of the grown crystal [14,15,16,17].

Aside from the quality of the seed, there are several parameters determining the quality of the resulting crystal. One important parameter is the axial temperature gradient present in the seed during heat up and the resulting elastic deformation before growth starts. Shioura et al. [18] proposed a model explaining the effect of BPD networks appearing in the seed after growth. Furthermore, nitrogen doping influences the lattice spacing and also the coefficient of thermal expansion (CTE) [19,20]. An inhomogeneous-doping level in the crystal should therefore induce stress during cooldown.

Furthermore, the mounting method of the seed will impact the stress present during heat up and cooldown. One possibility of seed attachment is utilizing a graphite plate with a certain thickness combined with a heat-resistant, carbon-based glue. This prevents the occurrence of thermal decomposition cavities, but it will also introduce stress due to the differing CTEs of carbon and SiC [21]. It is also possible to realize a thin carbon-based protection layer on the back of the seed to avoid this issue altogether.

Computed tomography using X-rays can be applied to monitor the growing crystal and the morphology change of the powder source during and after the growth process. While this allows optimizations regarding the growth kinetics of PVT, other process parameters cannot be evaluated [22,23].

For this reason, numerical modeling is used to further characterize the hot zone during and after crystal growth, considering temperature fields of the crystal and source powder [24,25,26], mass transport [27,28,29], stress and growth kinetics [30,31], or dislocation dynamics [32,33]. However, to the knowledge of the authors, the impact of the change in CTE caused by nitrogen doping on stress was not yet investigated numerically.

The aim of this work is to investigate the impact of the attachment method and also the varying doping levels in the initial seeding phase of a SiC crystal growth run on the types and distribution of defects close to the interface between seed and grown crystal. For that purpose, four 100 mm diameter 4H-SiC single crystals were grown employing the PVT method. During two crystal growth processes, the seed was fixed to a 7 mm graphite plate, the third crystal growth was performed with a seed mounted to a 1 mm graphite plate, and the fourth run was realized with a seed exhibiting a back-side coating. In addition, the nitrogen gas flux during the growth phases of the four crystals was varied. The bulk growth of crystal A was carried out with a constant nitrogen gas flux during the growth phase, whereas the three following growth runs exhibited an improved gradual nitrogen gas flux during the seeding phase.

Wafers containing the initial seeding phase were cut from the crystals, polished, and investigated by Raman spectroscopy to obtain doping levels and residual stress present in the material. Raman spectroscopy is a powerful tool as a non-destructive characterization method [34]. To further investigate the type of defects occurring during the initial-seeding phase, the samples were KOH-etched [35]. The doping levels obtained with Raman spectroscopy were implemented into a numerical model to predict the stress caused by the changed CTE during the cooldown phase.

## 2. Materials and Methods

In Table 1, the four PVT-grown 100 mm crystals, A–D, and their variations in growth parameters are depicted. The growth temperature was maintained between 2050–2150 °C measured at the crucible top, and the growth pressure was set between 5 and 30 mbar. After heating up to growth temperatures, the growth phase is initiated by decreasing the ambient pressure to the growth pressures mentioned previously. Before the ramp down of the ambient pressure started, the nitrogen gas flux was fixed to a constant 10% of the argon’s flux for crystal A. For crystals B–D, the nitrogen was ramped up during the decrease to growth pressures, reaching 5% of the argon’s flux in a linear fashion once growth has started. This ramp up intends to compensate for the additional nitrogen releasing from the isolations during the ramp down. The purities of the process gases are 4.8 and 5.0 for argon and nitrogen, respectively. All four crystals were grown utilizing a 4° off-oriented seed towards <11$\bar{2}$0>. To remove any organic residue, the 100 mm seeds are cleaned with technical acetone and ethanol. The seed-mounting type varied between an attached 7 mm graphite plate for crystals A and B, a thinner graphite plate of 1 mm thickness (crystal C) and a backside coating of the seed itself for crystal D without any graphite plate.

After the growth, all four crystals were sliced, polished, and characterized with Raman spectroscopy. The wafering was done in such a way that the transition between the seed and the grown crystal is visible on the Si-face of the wafer, also seen in Figure 1. The offcut angle of the samples varied between 0.5° and 1.4°. This way, the KOH etching can be used to further characterize defects lying in the interface layer. The Raman measurements were carried out with a Horiba LabRAM HR Evolution confocal Raman microscope. All measurements were performed utilizing a diode-pumped solid-state laser with a wavelength of 532 nm for excitation. The magnification was set to 10× and the grating to 1800 gr/mm. A line scan of spectra was taken from 160 cm^−1^ to 1100 cm^−1^ and evaluated with respect to the folded transverse optical (FTO) mode’s peak position at around 777 cm^−1^. The position of the line scan can be seen in Figure 1c. Furthermore, the longitudinal optical phonon–plasmon coupled (LOPC) mode’s peak position located at approximately 983 cm^−1^ was assessed. While the latter one exhibits an asymmetrical shape, its peak position can be used for an approximation of the present nitrogen doping [36,37]. Furthermore, Sugiyama et al. and Batten et al. reported a FTA peak position shift of −1.96 cm^−1^ per GPa of present stress, where a downshift corresponds to tensile stress and vice versa [38,39]. This relationship was utilized to approximate the stress between the seed and the grown crystal in a relative manner.

Comparing Raman measurements taken with different devices can lead to errors since every device has to be calibrated meticulously. To avoid possible shifts of the Raman measurements with respect to values of literature and to verify the calculated doping concentrations, Hall and Raman measurements were carried out on reference samples.

Using the values obtained from the Raman measurements, a numerical simulation was set up with the use of COMSOL Multiphysics. The observed doping gradient was implemented into a model of the PVT hot zone setup. The CTE of the doped SiC material was adjusted according to the doping level using the values from Stockmeier et al. [19]. However, it has to be noted that the authors only measured CTE values of undoped 4H-SiC and the CTE of 6H-SiC with varying dopants. Since the difference between the CTE value of undoped 4H and 6H-SiC remains constant over a wide range of temperatures, the same case was assumed for the CTE of high-doped 6H-SiC. The temperature dependent CTE of 4H-SiC was extrapolated by adding the difference to the CTE of high-doped 6H-SiC mentioned before. The semiempirical, quasi-harmonic model proposed by Reeber [40] was utilized with the fitting parameters for undoped 4H-SiC taken from [19]:(1)$$\alpha =\sum X_{n}\frac{{\left(\Theta_{n}/T\right)}^{2}\exp\left(\Theta_{n}/T\right)}{{\left(\exp\left(\Theta_{n}/T\right)-1\right)}^{2}},$$
where *T* corresponds to the temperature, *α* is the CTE, and *Θ_n_* and *X_n_* are the fitting parameters. For doped 4H-SiC (2 × 10^18^ cm^−3^), they are as follows: *Θ*_1_ = 745.8 K, *Θ*_2_ = 800.0 K, *Θ*_3_ = 1863.8 K, *X*_1_ = 287.7 × 10^−7^ K, *X*_2_ = −256.8 × 10^−7^ K, and *X*_3_ = 25.7 × 10^−7^ K. The relationship of the difference between the CTEs of undoped 4H-SiC and doped 4H-SiC at a specific temperature was assumed to be linear. For example, assuming the difference of the CTE between undoped 4H-SiC and 4H-SiC doped with 2 × 10^18^ cm^−3^ is 0.2 × 10^−6^ K^−1^, the CTE difference of 4H-SiC doped with 1 × 10^18^ cm^−3^ is expected to be 0.1 × 10^−6^ K^−1^.

Finally, the samples are etched with molten KOH of 85% purity to reveal the defect types and densities utilizing the FAU in-house KOH-etching setup. To be more specific, the samples are lowered into the etching chamber containing the molten KOH and etched for 5–8 min at 500–520 °C. The temperature was measured in situ using a thermocouple next to the sample holder. The sample of crystal A was etched for 5 min at 520 °C, while the sample of crystal B was etched for 5 min at only 500 °C. This was due to some process irregularities in the KOH-etching setup and resulted in a slightly smaller etch pit size. The samples of crystal C and D were etched at 510 °C for 8 min. After the etching process, the samples are cleaned with HCl to remove any residue of KOH. A more thorough description of the KOH-etching process is described by Sakwe et al. [35].

## 3. Results

### 3.1. KOH Etching

Figure 2 shows microscopic images taken from the etched Si-surface of the samples. The interface between the seed and the grown crystal is clearly visible. In Figure 2a, the initial seeding phase of crystal A is depicted with the seed area on the left and the grown crystal on the right. Both the seed area and the area of the grown crystal exhibit high amounts of defects arranged in a dislocation pattern similar to the observations of Shioura et al. [18]. The seed area is mainly covered by BPDs, while the area of the grown crystal is defined by stacking faults (SF), which in turn are restricted by more accumulated defects of presumed threading character (see inset and Figure 3f). The dislocation density of the seed crystal is determined as 1.18 ± 0.19 × 10^5^ cm^−2^. Figure 3 depicts magnified areas of crystal A’s interface. Figure 3e shows the size difference between BPDs and SFs. In Figure 3a, the seed–crystal interface is clearly visible, divided by a row of partial dislocations. The size difference between the etch pits of BPDs and partial dislocations can be seen in Figure 3e. Another indication that these smaller etch pits are SFs can be seen in Figure 3d, where an SF is clearly bound by two etch pits of partial dislocations. As one moves further into the crystal (Figure 3b,c), the etch pits of the partial dislocations increase in size, indicating that most of the SFs are generated in the very first layers of crystal growth. Figure 3f shows SFs interacting with a dislocation pattern consisting of threading dislocations. It is clearly demonstrated that the high nitrogen concentration leads to SF generation in the grown crystal since we cannot observe similar behaviors in crystals B–D with the optimized nitrogen doping.

The interface of crystal B illustrated in Figure 2b is expressed by a significant change in the defect density and types. This wafer was etched for a slightly shorter time; therefore, the etch pit sizes are smaller compared with crystal A. The seed crystal is still dominated by the BPD defect type (see inset), although with a much lower density of 1.22 ± 0.24 × 10^4^ cm^−2^, one magnitude lower than in crystal A. In addition, the interface is much more sharply defined than in Figure 2a. Microscopic images with a higher magnification are shown in Figure 4. There are no SFs apparent in the images; however, the grown crystal is characterized by a very high density of threading dislocations. The defect density decreases further into the crystal, as seen in Figure 4c. This decrease can also be observed in the sample of crystal A. In contrast to crystal A, there are almost no BPDs visible, neither next to the interface or farther in the bulk crystal. The improved nitrogen ramp up apparently had the effect of decreasing the defect density inside the seed; however, the interface is still very clearly defined in the sample. This is most likely caused by the mismatch between seed and grown crystal due to the utilized seed attachment method, which needs to be compensated in the beginning of the growth. While less apparent than in crystal A, there are still dislocation patterns present in crystal B, marked in Figure 4b.

Figure 2c shows the interface of crystal C grown with the same nitrogen gas flux but attached to a 1 mm instead of 7 mm graphite plate. BPDs are again the dominant defect type in the seed; the total dislocation density on the seed is 6.33 ± 1.87 × 10^4^ cm^−2^. Despite the thinner graphite plate and the therefore assumed lower stress during heating up and cooling down, the defect density is higher than the one of crystal B grown with a thicker graphite plate. It must be said that the interface on this sample is positioned next to the edge of the wafer and, therefore, could exhibit a higher defect density compared to the center of the wafer. The interface itself is exhibiting a high amount of threading dislocations in a similar manner to crystal B. While the graphite plate is substantially thinner than 7 mm, the numerical calculation in the following section confirms a similar amount of stress during the heat-up phase of the crystal growth run, therefore, introducing a comparable misfit.

The interface of crystal D depicted in Figure 2d is defined by a very smooth transition between the seed and the grown crystal. The observed BPD density in the seed area amounts to 9.24 ± 5.56 × 10^3^ cm^−2^. Unlike the previous wafers, there is no immediate increase of defects detected. Instead, two almost parallel arrays of BPD line-ups mark the initial seeding phase, indicated by the arrows. The backside coating instead of a graphite plate prevents a buildup of stress during the heat-up-phase. Therefore, only the misfit caused by the positive axial temperature gradient has to be overcome, resulting in a dramatic decrease in defect generation. This is further supported by the results of the numerical calculations, as explained in Section 3.

### 3.2. Raman Measurements

#### 3.2.1. Characterization of the LOPC Peak Positions

Based on the Hall measurements (not shown here), for our Raman setup the frequency of LO phonons *ω_L_* of undoped 4H-SiC was determined to be 963.46 cm^−1^. Similar values are reported in literature [34,36,37]. Figure 5 shows the calculated free carrier concentration based on the FLO peak going across the seed–crystal interface of crystals A and B. The x-axis values are converted into the distance perpendicular to the initial-seeding phase with the help of the sample’s determined off-axis cut angle. Figure 5a illustrates a sharp increase in free carrier concentration from 7.5 × 10^17^ cm^−3^ to a maximum of 6.7 × 10^18^ cm^−3^ once the grown crystal material is evaluated. It is known that nitrogen doping will increase during the initial phase of crystal growth if no appropriate measures are taken. This can be caused by residual nitrogen present in the isolation material, which will experience desorption during the initial pressure decrease. Lowering ambient pressure to start crystal growth will also increase isolation performance and, therefore, increase the temperature in the hot zone. A higher temperature has been shown to decrease nitrogen incorporation into SiC [41,42]. Furthermore, a slower growth rate is also connected to an enhanced dopant incorporation [43]. Therefore, the growth conditions in the initial stage, meaning a slow growth rate due to still comparatively high ambient pressure in combination with relatively low growth temperatures, result in a high-doping concentration in the first phase of growth despite a constant supply of dopant gas. Since crystal A was grown using a constant nitrogen flux of 10% of the argon’s gas flux during the entire growth run, the results of the Raman spectra depicted in Figure 5a are as expected.

For the growth of crystals B–D, adjustments have been made to prevent the doping variation. The nitrogen gas flux was changed from a constant value to the ramp up described in Section 2. In addition, the applied power during the pressure ramp up in the beginning stage of the growth was changed to accommodate the changing isolation properties and facilitate a constant temperature throughout the initial ambient pressure ramp up. The effects of these modifications on the growth process are depicted in Figure 5b. The free carrier concentration increases by about 60% from 2.2 × 10^17^ cm^−3^ to a maximum of 3.6 × 10^18^ cm^−3^, a significant improvement compared to crystal A exhibiting a doping level increase at the growth interface of ca. 800%, respectively.

#### 3.2.2. Characterization of the FTO Peak Positions

Figure 6 shows the calculated stress from the peak positions of the LOPC peak going across the seed–crystal interface of crystals A to D. The reference value for stress-free 4H-SiC material is 777 cm^−1^. Both crystals A and B depicted in Figure 6a,b were grown using a 7 mm graphite plate for the seed attachment. Crystal A exhibits a significant variation in stress, approaching a total difference of 1.41 GPa, while the difference for crystal B only amounts to 0.28 GPa. In all likelihood, the reduction of 1.13 GPa is caused by the lowered variation in nitrogen doping. Figure 6c,d illustrate the stress present in the samples of crystals C and D, both grown with the improved nitrogen flux ramp up. The total differences of 0.11 GPa and 0.20 GPa for crystal C and D, respectively, are slightly lower but of a similar magnitude as the stress observed in Figure 6b. However, it is outlined that the interface between seed and grown crystal in the wafer depicted in Figure 6c was close to the edge of the wafer. Therefore, the stress measurements could be affected by the fact that only a very thin slice of seed material was evaluated during the Raman measurements along with the crystal grown underneath. Since the interface was evident during the KOH etching, the results of the sample will still be presented for the sake of completeness. The small difference between the stresses observed in crystals B–D, despite the different mounting techniques, are surprising but can be explained by the numerical model shown in Section 3.

### 3.3. Numerical Results

#### 3.3.1. Stress Simulation of Differing Attachment Strategies

To determine the influence of the different seed-mounting strategies, a numerical model was set up. This model considers a hot zone setup at growth temperature, including the axial and radial temperature gradients used during growth and a crystal of 21 mm length in total and 100 mm in diameter. The seed is included in the total length of the crystal and amounts to 1 mm. The seed was either fixed to a 7 mm graphite plate, a 1 mm graphite plate, or without any graphite plate at all. It was assumed that the crystal plus graphite stack (if applicable) is completely stress-free at the growth temperature and has a uniform doping level of 3 × 10^18^ cm^−3^. This stack is then cooled down to room temperature. The r-component of the stress induced by the different CTE values of the materials can be seen in Figure 7. Figure 7a–c illustrate the distribution of stress next to the graphite plate. It is obvious that the thickest graphite plate also results in the highest amount of stress (to a maximum of −54.2 MPa); however, with −42.1 MPa, the 1 mm graphite plate induces only slightly less stress into the completely cooled crystal. The crystal grown with a backside coating experiences a maximum of −12.0 MPa, exclusively caused by the established temperature gradients. These stress values are much lower than the values measured with Raman spectroscopy in the previous section. Therefore, the seed attachment to graphite and the axial/radial temperature gradients cannot satisfactorily explain the high stresses observed in crystal A. In addition, the significant drop of 1.13 GPa of stress in crystal B, despite the similar growth parameters regarding temperature field and seed mounting in regard to crystal A, indicate that the nitrogen flux change should be the cause for this difference.

#### 3.3.2. Stress Simulation of Differing Nitrogen Doping

To investigate the influence of the changing CTEs caused by nitrogen on the stress prevalent in and around the transition between seed and crystal, the nitrogen dopant levels measured with Raman spectroscopy were implemented into the numerical model and the CTE was adjusted according to Stockmeier et al. [19] and the adaptions mentioned in Section 2. Apart from this, the same boundary conditions apply as in the previous section, e.g., stress-free state of a 21 mm crystal at growth temperature. In Figure 8a,b, the r-component of the stress is shown for crystals A and B, respectively. The data is presented in a similar way as the samples were prepared, e.g., a slice through the crystal almost parallel to the basal plane while cutting the interface between the seed and the grown crystal. This was done to showcase the thin area of high doping in the case of crystal A. Figure 8c,d depicts the r-component of stress plotted over the distance of the interface. The calculated stress inside the highly doped initial-seeding layer of crystal A is one magnitude bigger than the stress derived from the uniformly doped case presented in Figure 7a,d. The seed in crystal A experiences a compressive stress of 854.4 MPa, neighboring the highly-doped grown crystal layer exhibiting a tensile stress of 817.4 MPa. This difference of 1.67 GPa can explain the high stress measured in crystal A. In addition, it explains the big discrepancy between crystals A and B regarding the observed stress, despite the fact that both crystals were grown on a 7 mm graphite plate. The stress induced by the mismatch of the CTE values between the graphite plate and the seed is two orders of magnitude lower than the stress caused by a varying doping concentration in the range present in crystal A. Moreover, the jump of the compressive stress in the seed to the tensile stress in the grown crystal observed in Figure 6a is also present in the simulation of crystal A. However, the sharp decline of stress going further into the grown crystal area as predicted in the numerical calculations is not seen in the conducted Raman measurements. It is likely that the sharp peak of stress as predicted by the simulation will be relaxed during the high-temperature cool-down phase after growth has finished. This coincides with the observations made in the KOH etchings. The difference of 1.67 GPa describes the high-stress edge case since only elastic and no plastic deformation is assumed. The other edge case would be complete relaxation via plastic deformation above the reported brittle–ductile transition temperature of SiC and exclusively elastic deformation below this temperature [44]. If the numerical calculations are repeated with a transition temperature of 1100 °C, e.g., cooling down the crystal from 1100 °C to room temperature rather than cooling down from growth temperatures, we obtain a difference of 0.99 GPa between the seed and the grown crystal. The real value will lie in between these two values, depending on how much stress was relieved during the initial cooldown phase.

## 4. Discussion

The microscopic images on the KOH-etched interfaces in Figure 2 and Figure 3 clearly show the difference caused by the changed doping strategy. The difference in CTE of graphite and seed during the heat up leads to a concave shape of the graphite-seed stack. If the BPDs observed in the seed area would have been mainly caused by a relaxation of the strained seed before growth starts, the defect density of crystals A and B in the seed should be similar. However, we observe the difference by comparing the vastly different BPD density of crystal A (1.18 ± 0.19 × 10^5^ cm^−2^) to the one of crystal B (1.22 ± 0.24 × 10^4^ cm^−2^). Numerical simulations of the seed mounted onto the graphite plate confirm the concave shape in the beginning of the growth. If the lattice spacing of the strained seed-graphite stack is compared with the lattice spacing of strain-free and similarly-doped SiC at growth temperatures, we can approximate the needed dislocation density for a complete relaxation of strain. The derived density of around 10^9^ cm^−2^ is very high but qualitatively supports the etch pit density found in the area of the newly grown crystals B and C. However, for an exact determination of the etch pit density, KOH etching is not suitable since in our samples the amount of etch pits leads to overlapping (see Figure 4a).

It has been shown in literature that the start of SiC PVT growth is characterized by many separate growth islands, merging step by step with progressing growth time [45]. We can observe similar networks in crystal A, consisting of threading dislocations enclosing SFs and reducing the misfit between the seed and the first few epilayers. Due to the high nitrogen concentration variation of crystal A during the cool-down phase, a very high stress is imposed on the crystal, promoting dislocation glide. BPDs and SFs will be activated before threading dislocations due to the higher required energy necessary for the prismatic slip system [46]. These will be mobile until they encounter the walls of threading dislocations, as seen in Figure 3f. This could result in the observed dislocation networks in the growth layer of crystal A and the increased defect density in the seed compared to the crystals grown with the improved nitrogen-doping ramp up. 

It is noticeable that crystal A seems to compensate the initial misfit mainly with SFs, and crystals B and C are exhibiting almost exclusively TEDs. A possible explanation for the observed difference in defect types between the crystals A and B/C could lie in the impact of nitrogen on the lattice spacing of highly doped SiC. It is known that nitrogen doping leads to SF incorporation in the SiC growth [43]. Furthermore, the seed-graphite stack experiences compressive stress after reaching the growth temperature (see Figure 9a). Since the lattice spacing decreases with increasing nitrogen and crystal A was grown with a very high nitrogen-doping concentration, the mismatch is reduced compared to the mismatch in the beginning of the growth of crystals B and C [20,47]. This reduction in mismatch and, therefore, strain could have been a reason for the appearance of SFs instead of TEDs.

Since crystals B and C were grown utilizing a more homogeneous nitrogen distribution, the samples experience a lower amount of stress during the cooldown phase. Therefore, dislocation glide at the interface is diminished and mainly the threading dislocations caused by the initially concave seed-graphite stack can be observed. One possible explanation for the observed TEDs in the interface can be seen in Figure 9. In the beginning of growth, a misfit exists between the newly grown layer and the seed. This misfit could be accommodated by the generation of BPDs, which in turn get converted to TEDs. Interestingly, in the KOH-etched samples, no BPDs were observed next to the interface in the area of the grown crystal for crystals B and C, which would have been expected since BPD generation for misfit relaxation has been reported in the literature before [45]. Because of the very high dislocation density in the first layers of the grown crystal, it is possible that the etch pits of BPDs are obscured by the high amount of TEDs. Two possibilities of dislocation arrangements are shown in Figure 9b,c. If a BPD is generated to relieve the compressive stress present at the interface of the seed, it could be converted to a TED either at one or two ends. For a conclusive statement, X-ray topography should be utilized to observe the exact path the dislocation takes. It must be said that while the seed-graphite stack is concave, the temperature field is still convex with respect to the growth interface.

The results of crystal D show that by removing the main cause of the misfit (no graphite plate for seed mounting), the threading dislocations also disappear at the interface. We know from our numerical calculations that in our growth setup, a 1 mm seed with a backside-coating, instead of a graphite plate, only experiences a stress of 0.15 MPa before growth starts. This stress is caused by the axial and radial temperature gradients present in the seed. In this case, the model suggested by Shioura et al. fits nicely for the dislocations observed in the sample [18].

## 5. Conclusions

As a conclusion we could show that the nitrogen doping is the dominant cause for stress in the case of a highly inhomogeneous doping profile during the seeding phase. The magnitude of the stress was simulated numerically and validated by Raman measurements of the interface. The defects were characterized by KOH etching. The role of the mounting method was discussed and an explanation proposed for the observed threading dislocations at the interface of the crystals grown with a seed attached to a graphite plate. It was shown that a growth process using an adapted doping profile and mounting method will lead to an improved initial-seeding phase.

## Figures and Tables

**Figure 1 materials-15-01897-f001:**
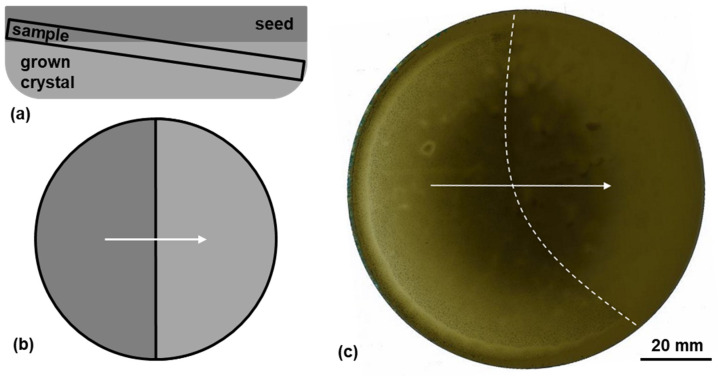
(**a**) Schematic of the sample’s position cut from the respective crystal; (**b**) schematic resulting sample; the arrow indicates the position of the line scan of the Raman measurements; (**c**) optical scan of the wafer cut from crystal A with the (0001)-side up; the arrow indicates the position of the line scan of the Raman measurements, the dashed line marks the transition between the seed (left side) and the grown crystal (right side).

**Figure 2 materials-15-01897-f002:**
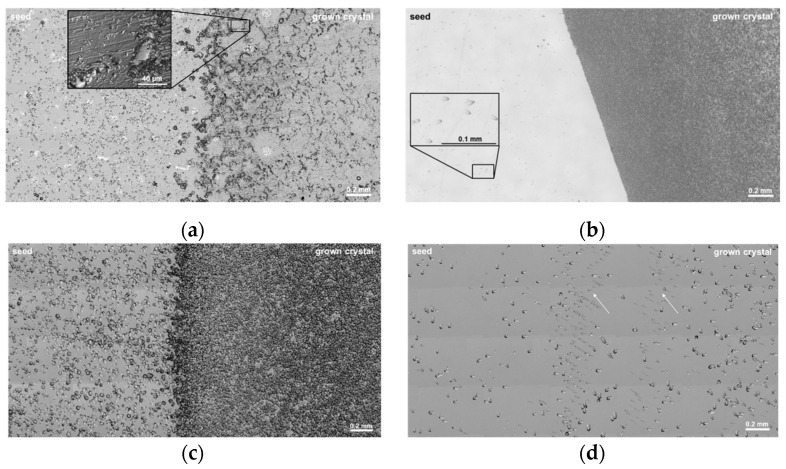
Microscopic images of KOH-etched wafers depicting the transitional layer between seed and grown crystal of (**a**) crystal A, (**b**) crystal B, (**c**) crystal C, and (**d**) crystal D.

**Figure 3 materials-15-01897-f003:**
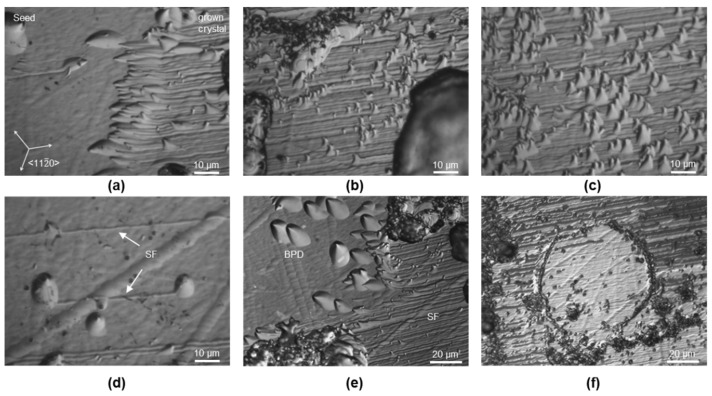
Microscopic images of the KOH-etched wafer taken from crystal A: (**a**) interface between seed and grown crystal; (**b**–**d**) images taken further to the right of the interface inside the area of the grown crystal; (**e**) two partial dislocations outlining a SF; (**f**) SFs interacting with dislocation pattern consisting of threading dislocations.

**Figure 4 materials-15-01897-f004:**
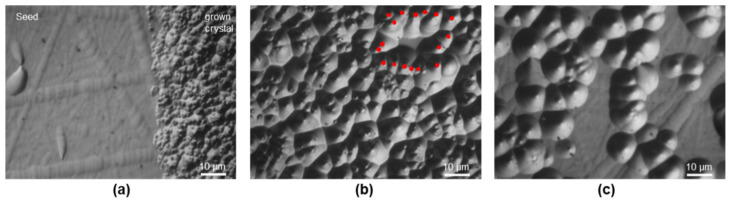
Microscopic images of the KOH etched wafer taken from crystal B: (**a**) interface between seed and grown crystal; (**b**,**c**) images taken further to the right of the interface inside the area of the grown crystal. A dislocation network is marked in (**b**).

**Figure 5 materials-15-01897-f005:**
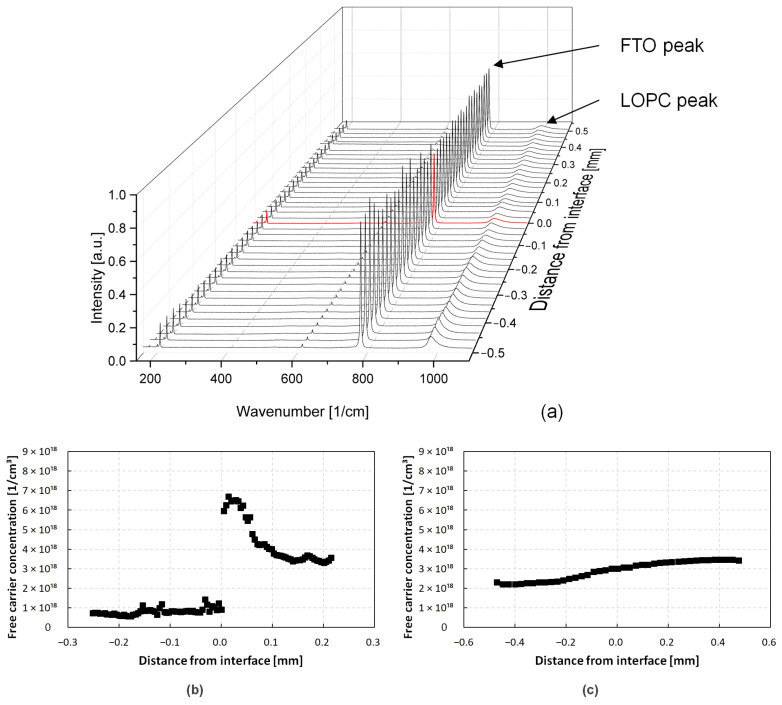
(**a**) Line scan of Raman spectra taken from crystal B; the position of the seed–crystal interface is marked in red; (**b**,**c**) measured free carrier concentrations across the seed–crystal interface of (**b**) crystal A and (**c**) crystal B, evaluated from the LOPC peak position. A positive distance from interface marks the area of grown crystal, while a negative distance indicates an area of the seed of the crystal.

**Figure 6 materials-15-01897-f006:**
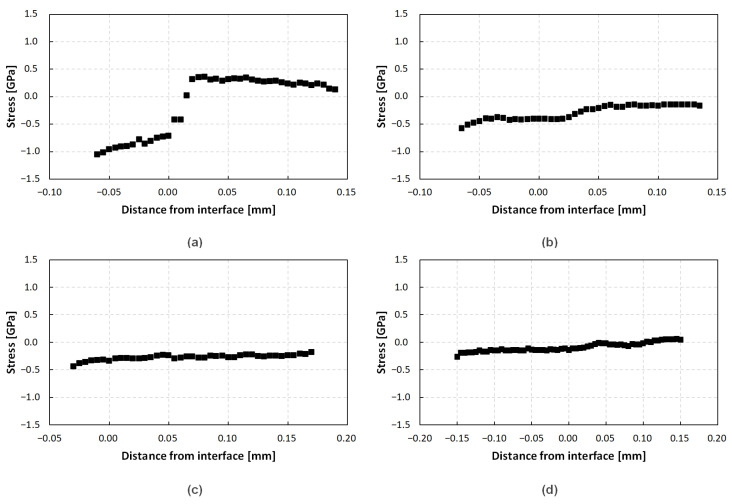
Measured stress across the seed–crystal interface of (**a**) crystal A, (**b**) crystal B, (**c**) crystal C, and (**d**) crystal D, evaluated from Raman measurements of the FTO peak position. The position of the line scan is depicted in Figure 1b,c. A positive distance from interface marks the area of grown crystal, while a negative distance indicates an area of the seed of the crystal.

**Figure 7 materials-15-01897-f007:**
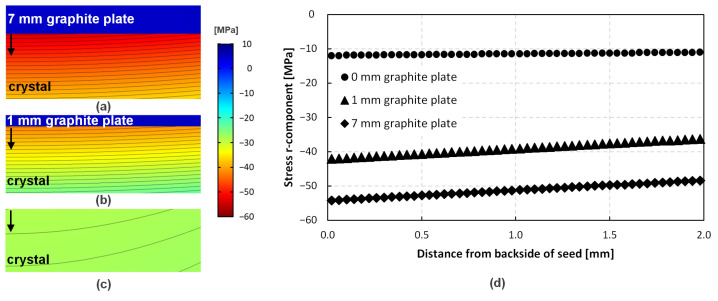
Numerical simulation depicting the r-component stress field of a 21 mm crystal cooled down to room temperature from growth temperature, attached to a (**a**) 7 mm graphite plate; (**b**) 1 mm graphite plate; (**c**) with a carbon-based backside coating. (**d**) illustrates the dataset taken at the positions of the arrows in (**a**–**c**).

**Figure 8 materials-15-01897-f008:**
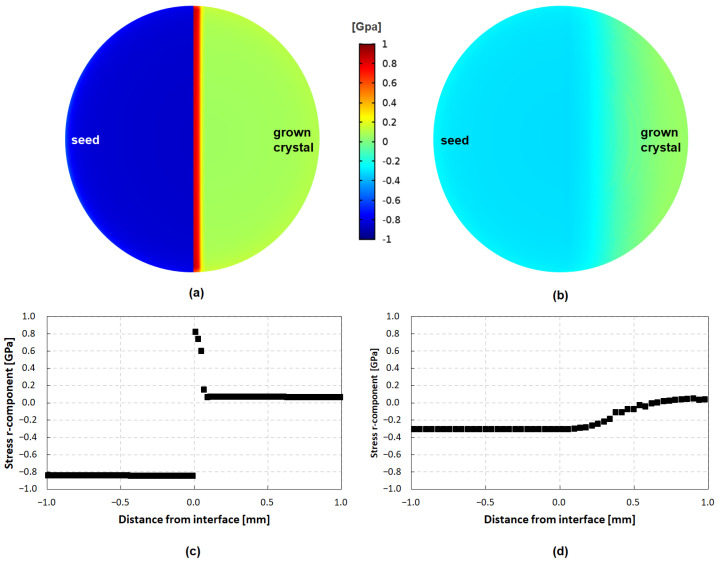
Calculated r-component stress across the seed–crystal interface of (**a**) crystal A and (**b**) crystal B, calculated from Raman measurements and considering the change in CTE caused by changing doping concentrations. A positive distance from interface marks the area of grown crystal, while a negative distance marks the seed portion of the crystal. The plotted r-component over the distance from the interface is illustrated for crystal A and crystal B in (**c**,**d**), respectively. A positive value indicates a tensile stress while compressive stress is illustrated by positive values.

**Figure 9 materials-15-01897-f009:**
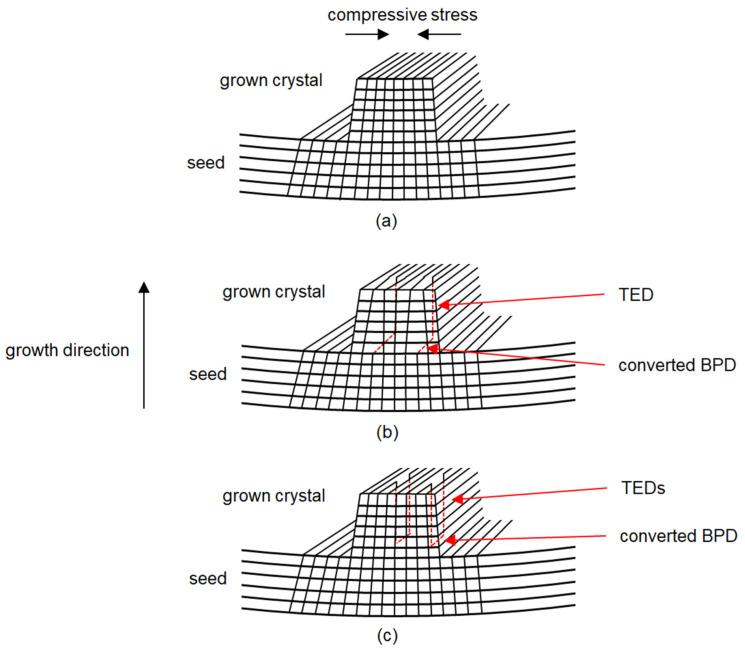
(**a**) Schematic illustration of compressive stress present after heat-up to growth temperatures due to the seed mounted on a graphite plate; (**b**) BPD converted into a TED; (**c**) BPD converted into two BPDs.

**Table 1 materials-15-01897-t001:** Overview of the grown crystals and the variations of their growth parameters.

Grown Crystals	Mounting Method	N_2_-Flux in % of Ar-Flux
Crystal A	7 mm graphite plate	constant 10%
Crystal B	7 mm graphite plate	ramp up to 5%
Crystal C	1 mm graphite plate	ramp up to 5%
Crystal D	backside coating	ramp up to 5%

## Data Availability

Not applicable.

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
