# Peer review of "Impact of Mechanical Stress and Nitrogen Doping on the Defect Distribution in the Initial Stage of the 4H-SiC PVT Growth Process"

_materials, 2022, doi:10.3390/ma15051897_

Round 1
Reviewer 1 Report
The manuscript focuses on investigation the effect of N2 doping concentration on defect distribution in SiC crystals during PVT growing. The manuscript will be interesting for the readership of the Materials, however some important experimental details about seeding phase and etching procedure is missing.
There are some concerns that need to be addressed:
1) Please indicate the purity and suppliers for all the chemicals and gases that were used in the study.
2) The details about seeding phase preparation ad specification is missing. The conditions for KOH etching should be specified.
3) 11 line 409 – the reference is missing.
Reviewer 2 Report
The paper describes the influence of mechanical stress and doping on the defct distribution in 4H-SiC grown with the PVT technique. The manuscript is usually clear and well organized but an English polishing is necessary and needs major revision work to be suitable for publication in Materials. In the following my detailed comments:
- In the manuscript there is a recurrent use of words such as attach, attachment, run …. which undermine the text fluency. Please consider the use of synonyms.
- Authors have reported several times in the text type and characteristic of seeding. Please limit the repetition to the necessary.
- Line 62. Please modify the sentence “… nitrogen doping levels were varied between the four crystals.” . It is clear what Authors mean but it is not clear in the text. Between should be changed in among, and I imagine that the doping has been changed during the growth and not in the final crystals.
- In many part of the text Authors claim the growth of 100 mm crystals, but it is never specified if 100 mm is the diameter, the length or whatever.
- Lines 79-82. Please make an effort to make these two sentences more linear and clear.
- Lines 95-96. Please use a proper terminology. The sentence sounds terrible.
- Line 104. Again use a proper terminology. A wavelength shift toward higher wavelengths is a redshift and not a downshift.
- Section 3.2 Raman measurements. In my opinion at least one Raman spectrum should be reported. Figures 5 and 6 are obtained from Raman spectra and one example should be shown.
- Figures 5, 6 and 8 (c,d) are of poor quality for the production. Labels are very hard to read while a lot of space has been left free.
After these revisions, I could recommend the publication of this manuscript in Materials.
Reviewer 3 Report
This work reveals the impact of nitrogen doping concentration on stress in the PVT grown 4H-SiC. It is suggested that coefficient of thermal expansion is the dominating factor for types and density of defects during PVT
growth of 4H-SiC. Moreover, compared to munting method, inhomogeneous nitrogen doping leads to higher overall stress.
Overall work looks good and will be of interest to the scientific community working on SiC crystal growth. However, there are some
ambiguities, as given below, which need to be addressed before its acceptance.
1. Please check thoroughly for typos, e.g.,
line 166, full stop is missing
line 190, "chapter" should be replaced by work/manuscript.
2. In table 1, how the choice of nitrogen flux was made?
3. Fonts in figure 5, 6 and 8 are hard to see. Please increase their size.
Reviewer 4 Report
File attached

Round 2
Reviewer 1 Report
I am satisfied with the answers of the authors and the changes made in the revised version.
I have just a few suggestions:
1) Could you please update the quality up to 300 dpi of all graphics in the manuscript.
2) I suggest introduce a color coding of different graph on Figure 5a.